# Development of Stereo Visual Odometry Based on Photogrammetric Feature Optimization

## Sung-Joo Yoon and Taejung Kim *

Department of Geoinformatic Engineering, Inha University, 100 Inharo, Michuhol-gu, Incheon 22212, Korea; 22181415@inha.edu

\* Correspondence: tezid@inha.ac.kr; Tel.: +82-32-860-7606

**Abstract:** One of the important image processing technologies is visual odometry (VO) technology. VO estimates platform motion through a sequence of images. VO is of interest in the virtual reality (VR) industry as well as the automobile industry because the construction cost is low. In this study, we developed stereo visual odometry (SVO) based on photogrammetric geometric interpretation. The proposed method performed feature optimization and pose estimation through photogrammetric bundle adjustment. After corresponding the point extraction step, the feature optimization was carried out with photogrammetry-based and vision-based optimization. Then, absolute orientation was performed for pose estimation through bundle adjustment. We used ten sequences provided by the Karlsruhe institute of technology and Toyota technological institute (KITTI) community. Through a two-step optimization process, we confirmed that the outliers, which were not removed by conventional outlier filters, were removed. We also were able to confirm the applicability of photogrammetric techniques to stereo visual odometry technology.

**Keywords:** stereo visual odometry; photogrammetry; feature optimization; pose estimation; KITTI

## 1. Introduction

Estimation of a platform's pose using a sensor is a technology that has attracted attention in various fields, such as robotics and the automobile industry. Typical sensors include the global positioning system (GPS), light detection and ranging (LiDAR), and the camera. The GPS is the most popular method, and sub-meter accuracy is possible. However, accurate GPS equipment is very expensive, and accuracy is greatly reduced in some environments where satellite signals are blocked, such as downtown or in tunnels [1]. The method using LiDAR is very accurate and stable. However, since it requires expensive equipment, its application is limited. The method using a camera has a great advantage that the construction cost is relatively low. This technique is called visual odometry (VO). VO is divided into monocular visual odometry (MVO) and stereo visual odometry (SVO). The MVO is slightly cheaper because it uses one camera, but there is a scale uncertainty problem in pose estimation [2]. It also has relatively unstable image geometry [3]. SVO has an advantage that camera localization and generation of 3D maps around the vehicle can be achieved simultaneously. For both MVO and SVO, accuracy and performances are highly dependent on the image processing algorithms applied. In this study, we focus on SVO.

There has been a significant amount of research on SVO particularly focusing on how to extract its favorable features without any outliers. Stereo odometry algorithm relying on feature tracking (SOFT2) [4], which has been known to perform optimally, implemented simultaneous localization and mapping (SLAM) by performing pose estimation and mapping in parallel. It utilized blob and corner masks to extract features and the essential matrix to estimate the pose. It also considered the loop closing for feature and keyframe management. As features are extracted depending on the

rotation, there is a disadvantage that the performance may degrade depending on the state of the viewpoint. The RotRocc+ [5] method studied the characteristics of the optical flow and reprojection error for odometry and eliminated outliers by decoupling the optical flows of motion and exploiting the characteristics of the flow using a restrictive motion model. Therefore, there was a problem that outliers were not able to be accurately removed when the estimated vehicle's motion deviated from the model. Gradient-based direct visual odometry (GDVO) [6] method used a dual Jacobian optimization with a multiscale pyramid scheme for outlier removal. This method also applied gradient feature representation to respond to the lighting changes. However, it did not apply bundle adjustment, and therefore, the coordinates of the features were incorrect. Elbrus [7] applied the multiple pyramid Kanade–Lucas–Tomasi (KLT) method to track the feature and selected inliers using 2D track average motion and rate of disappearance. This method searched for features on multiple scales. It did not use depth information for elimination outliers. Circular fast retina keypoint (FREAK)-oriented fast and rotated binary robust independent elementary feature (ORB) visual odometry (CFORB) [8] detected features based on FREAK-ORB and repeated the process 50 times to perform random sample consensus (RANSAC) [9] for outlier elimination. This process was carried out using the concept of circular matching. Other methods used various ways to determine inliers [10–14]. It is notable that VO is based on features, and the accuracy of a method is highly related to the state of the feature. Most of the previous methods mentioned above optimized features based on pixels. In this study, we try to perform feature optimization using image geometry. Based on the photogrammetric analysis, we aim to apply image geometry for feature optimization and pose estimation.

This paper is structured as follows. Section 2 describes the material and proposed method. The experimental results are introduced in Section 3. Then, Section 4 shows a discussion of the results describing the strengths and weaknesses. Finally, Section 5 concludes.

## 2. Materials and Methods

For the experiment, we used the KITTI dataset provided by The KITTI vision benchmark suit [15]. The KITTI dataset was acquired with the vehicle shown in (a) of Figure 1, and these are distributed on the KITTI website as in (b) of Figure 1. It contains 11 image sequences and true values for the poses constructed in various environments, including urban areas, highways, and tree roads. The images were taken with an optical lens at a viewing angle of about 90 degrees. The camera used was a Sony ICX267 with a size of 1241 × 376 pixels. These were mounted on a rectified stereo rig.

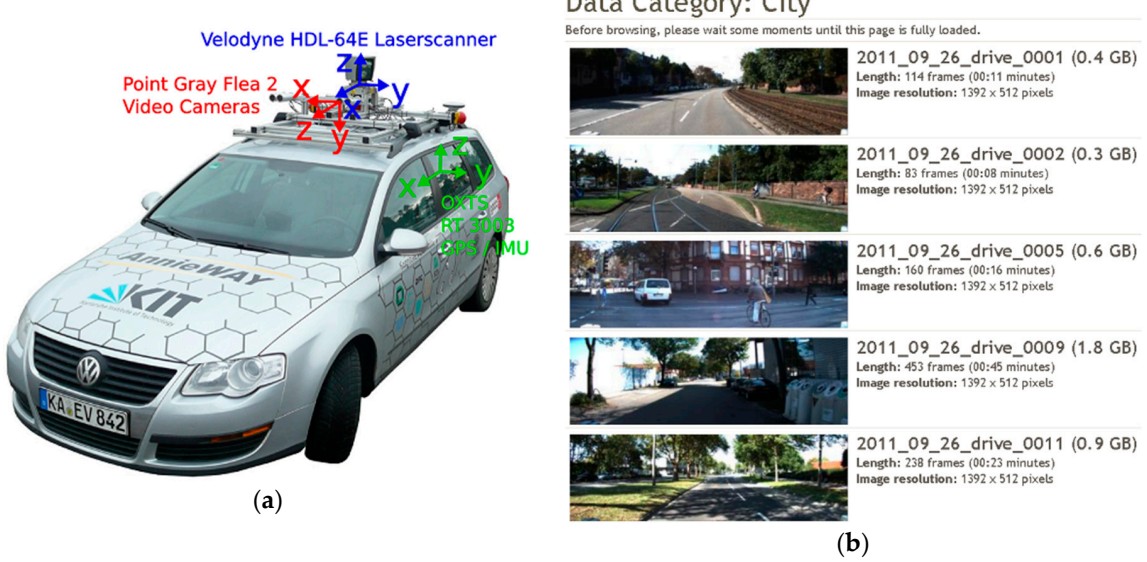

(a)

(b)

**Figure 1.** (**a**) Karlsruhe institute of technology and toyota technological institute (KITTI) platform; (**b**) KITTI dataset example [16].

The flowchart of the proposed real-time visual odometry technique is shown in Figure 2. First, we extracted features from images and searched for corresponding points by matching. This process is important because it takes a significant amount of time in the whole process, and the number of features and the matching result affect the accuracy of the estimation. Therefore, we compared the processing time and the number of corresponding points of several candidate methods. Next, we optimized the corresponding points. As mentioned, this process is necessary because the degree of the outlier affects the accuracy. In this study, we applied photogrammetry-based and computer vision-based optimization. In the photogrammetry-based part, we checked the reprojection error and the distance between the calculated and projected model points. This part is performed after the second frame because the geometry information between the previous and current image is needed. In the vision-based part, the outlier filtering in multiple images was based on RANSAC. Finally, we estimated the pose using the optimized corresponding points. It was based on the absolute orientation of the photogrammetric bundle adjustment. Finally, the relative positions of the platform were calculated by continuously accumulating the estimated pose. The detailed explanations are as follows.

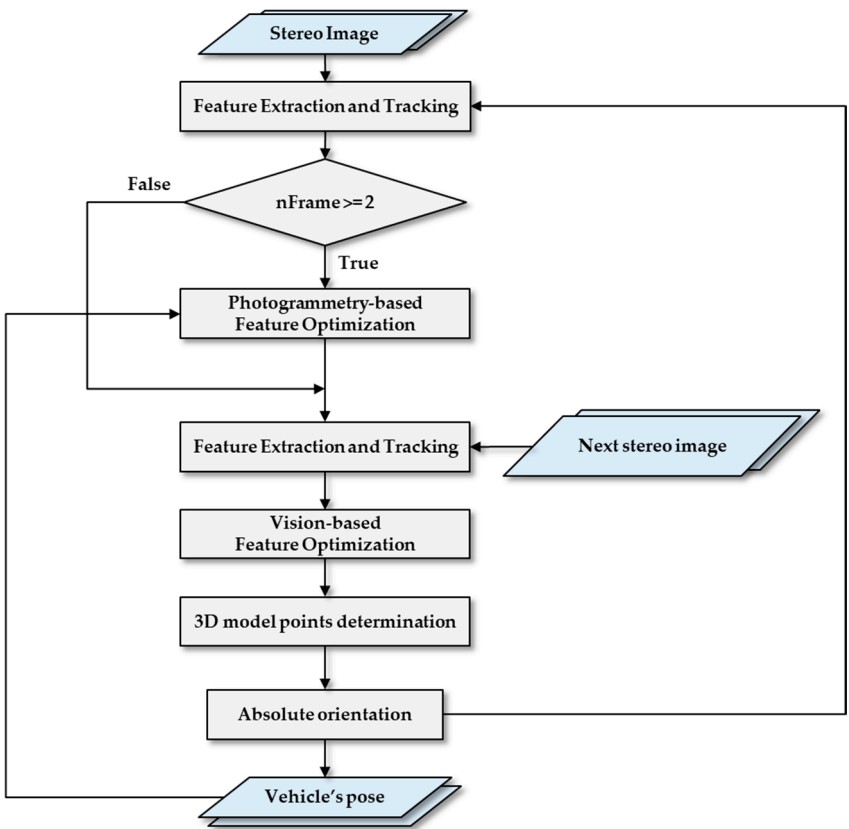

**Figure 2.** Procedure for proposed method.

## 2.1. Feature Extraction and Matching

The corresponding point extraction was carried out in the order of feature extraction and feature matching. First, in feature extraction, we extracted features such as the corner points or edges on the image. We selected representative feature extractors, scale invariant feature transform (SIFT) [17], speed-up robust feature (SURF) [18], and features from accelerated segment test (FAST) [19], Shi–Tomasi [20], provided by OpenCV.

Feature matching is divided into pairwise matching and sequential tracking. In pairwise matching, feature description, and matching are performed. After feature extraction, we calculated feature descriptors as shown in Table 1, and compared the resemblance to determine a corresponding point according to the matchers listed in Table 1. In sequential tracking, we set a window around a

feature of one image and tracked the corresponding point from the next image in an image sequence. Tables 1 and 2 summarize various techniques for pairwise matching and sequential tracking applied in this study.

**Table 1.** The pairwise matching methods tested.

| Detector | Descriptor | Matcher |
|---|---|---|
| Scale invariant feature transform (SIFT) | SIFT | Brute-Force |
| Speed-up robust feature (SURF) | SURF | Brute-Force |
| Features from accelerated segment test (FAST) | Binary robust invariant scalable keypoints (BRISK) | Fast library for approximate nearest neighbors (FLANN) |
| FAST | Oriented fast and rotated binary robust independent elementary feature (ORB) | FLANN |
| FAST | Fast retina keypoint (FREAK) | FLANN |

**Table 2.** The feature tracking method combinations.

| Extractor | Tracker |
|---|---|
| FAST | Kanade–Lucas–Tomasi tracker |
| Shi–Thomasi corner | Kanade–Lucas–Tomasi tracker |

*2.2. Corresponding Point Optimization*

In Figure 3, the green line indicates the feature movement direction between previous and current images. The red dot indicates the head direction. In the process of feature extraction, points on moving objects such as cars can be extracted, as shown in the squares in Figure 3. These features have abnormal motion vectors and make camera motion misdiagnoses, which greatly reduce the accuracy of the overall process. The top image was taken while the vehicle was turning right. The motion vectors within the box were in the opposite direction to the others, which were not correct. The bottom image was taken at a constant speed. The large motion vectors within the box were also not correct. Based on these observations, we tried to remove the outlier based on the geometry.

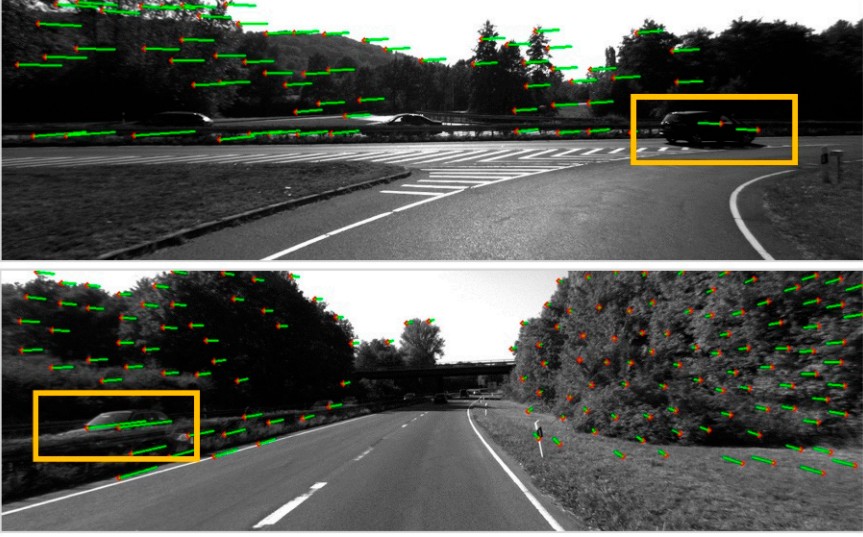

**Figure 3.** Example of moving objects in image. Top: sequence 01; frame 07. Bottom: sequence 01; frame 46.

Figure 4 shows the proposed feature optimization concept. As shown in the figure, we performed photogrammetry-based optimization using the previous and current images and vision-based optimization using the current and new images. The photogrammetric optimization was performed from the second image because it needed the image geometry.

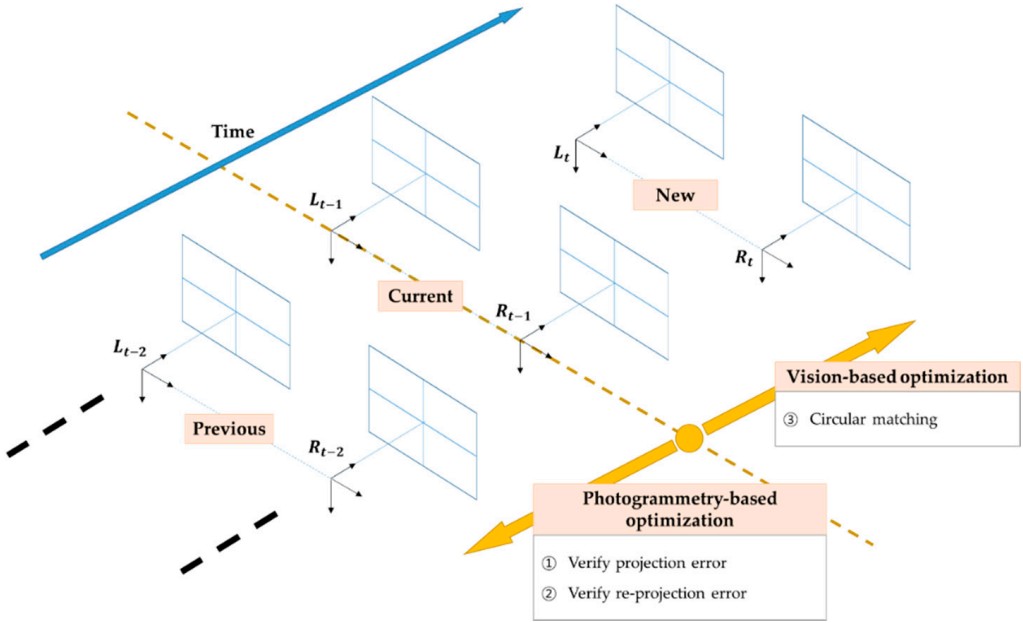

**Figure 4.** Feature optimization concept diagram.

Figure 5 explains photogrammetric optimization process in detail. Suppose that we have exterior orientation parameters estimated for the image pair ($L_{t-1}$ and $L_{t-2}$). The features on the previous images can be projected onto the current images ($L_{t-1}$ and $R_{t-1}$) through the estimated exterior orientation parameters (EOP). When the accurate image point is projected, the projected model point has a small separation from the calculated model point. Also, this model point is re-projected onto the previous image; it has a small separation from the corresponding image point. However, in the case of the inaccurate image point on the current image, when projecting or re-projecting it on the previous image, the differences are large. Based the separation distance, we selected optimized features.

$$\begin{bmatrix} Prj_X \\ Prj_Y \\ Prj_Z \\ 1 \end{bmatrix} = \begin{bmatrix} r_{11} & r_{12} & r_{13} & T_x \\ r_{21} & r_{22} & r_{23} & T_y \\ r_{31} & r_{32} & r_{33} & T_z \\ 0 & 0 & 0 & 1 \end{bmatrix} \begin{bmatrix} X_{t-1} \\ Y_{t-1} \\ Z_{t-1} \\ 1 \end{bmatrix}, \tag{1}$$

$$Distance(d_1) = \sqrt{(X_{t-2} - Prj_X)^2 + (Y_{t-2} - Prj_Y)^2 + (Z_{t-2} - Prj_Z)^2}, \tag{2}$$

$$\begin{cases} d_1 < threshold_1 : Status_1 = True \\ d_1 > threshold_1 : Status_2 = False \end{cases}, \tag{3}$$

where $X_{t-1}$, $Y_{t-1}$ and $Z_{t-1}$ are object coordinates in the ground coordinate system at time $(t-1)$. $T_x$, $T_y$, $T_z$ are translation elements from $L_{t-1}$ to $L_{t-2}$. $r_{11\sim33}$ are $3 \times 3$ rotation elements from $L_{t-1}$ to $L_{t-2}$.

$$\begin{aligned} \text{Reprj}_x &= -f\frac{r_{11}(X_{t-1}-T_x)+r_{12}(Y_{t-1}-T_y)+r_{13}(Z_{t-1}-T_z)}{r_{31}(X_{t-1}-T_x)+r_{32}(Y_{t-1}-T_y)+r_{33}(Z_{t-1}-T_z)} \\ \text{Reprj}_y &= -f\frac{r_{21}(X_{t-1}-T_x)+r_{22}(Y_{t-1}-T_y)+r_{23}(Z_{t-1}-T_z)}{r_{31}(X_{t-1}-T_x)+r_{32}(Y_{t-1}-T_y)+r_{33}(Z_{t-1}-T_z)}, \end{aligned} \tag{4}$$

$$Distance(d_2) = \sqrt{(x_{t-2} - \text{Repr}j_X)^2 + (y_{t-2} - \text{Repr}j_Y)^2}, \tag{5}$$

$$\begin{cases} d_2 < threshold_2 \ : Status_2 = True \\ d_2 > threshold_2 \ : Status_2 = False \end{cases}, \tag{6}$$

where $f$ is the focal length. $X_{t-1}$, $Y_{t-1}$ and $Z_{t-1}$ are the object coordinates in the ground coordinate system at time $(t-1)$. $T_x$, $T_y$, and $T_z$ are the translation elements from $L_{t-1}$ to $L_{t-2}$. $r_{11\sim33}$ are $3 \times 3$ rotation elements from $L_{t-1}$ to $L_{t-2}$. $x_{t-2}$, $y_{t-2}$ are image coordinates at time $(t-2)$.

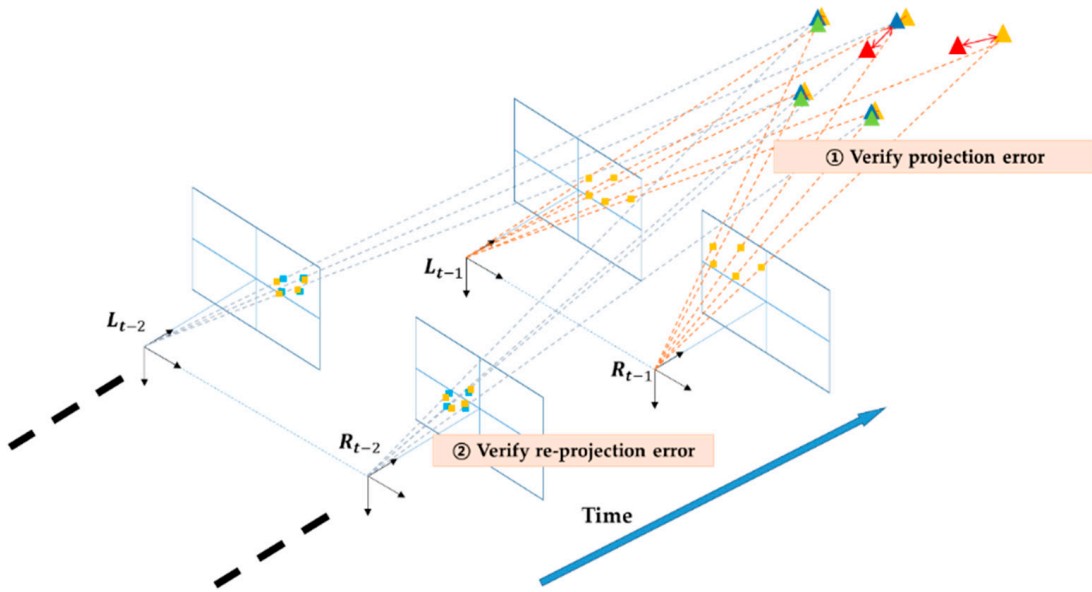

**Figure 5.** Corresponding point verification concept diagram.

In Equations (1) and (4), the translation and rotation elements are estimated on the pose estimation step. Through Equation (1), the model point on the model space at $(t-1)$ is projected onto the model space at $(t-2)$. Through Equation (2), the distance between the projected and actual model points is calculated on the model space. The distances are calculated for all features and classified as a threshold$_1$ as in ① of Figure 5.

Through Equation (4), the model point on the model space at $(t-1)$ is re-projected onto the image at $(t-2)$. Through Equation (5), the distances between the re-projected and the actual image points are calculated on the image space and classified as a threshold$_2$ as in ② of Figure 5. As in Equations (3) and (6), if both threshold$_1$ and threshold$_2$ are satisfied, the feature is extracted as an inlier. Also, while checking the number of features, this process is repeated using the previous images.

In vision-based optimization, the RANSAC-based outlier filtering over multiple images was performed as in Figure 6. This method extracts random samples from the data and creates a model. Then, it selects the appropriate model while inputting the remaining data. In this process, outliers are removed. Within the next stereo pair ($L_t$ and $R_t$), we first extracted features corresponding to the features classified as inliers through photogrammetry-based optimization at $(t-1)$. Then, we eliminated the outliers by applying RANSAC while combining two images. We applied RANSAC in order from ① to ③ in Figure 6, and the features recognized as inliers in 4 images were saved for pose estimation.

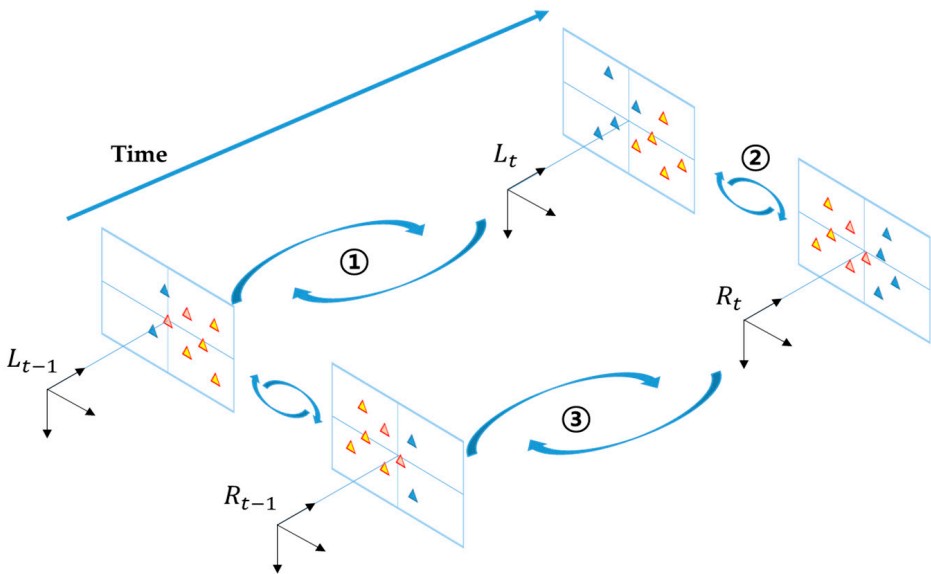

**Figure 6.** Multiple image filtering concept diagram.

### 2.3. Absolute Orientation for Pose Estimation

We estimated the platform's pose through absolute orientation using the collinearity condition. The collinearity condition is a condition that the three-dimensional coordinates of the object existing in the image, the image coordinates, and the camera projection center must be on the same straight line as shown in Figure 7. First, we determined the model points defined as $(P_n)$ between $O_1$ and $O_2$. Then, we established the relationship between $p_n$ and $P_n$ based on the collinearity equation as in Equation (7).

$$
\begin{aligned}
F &= x_n - f \frac{r_{11}(X_n - T_x) + r_{12}(Y_n - T_y) + r_{13}(Z_n - T_z)}{r_{31}(X_n - T_x) + r_{32}(Y_n - T_y) + r_{33}(Z_n - T_z)} \\
G &= y_n - f \frac{r_{21}(X_n - T_x) + r_{22}(Y_n - T_y) + r_{23}(Z_n - T_z)}{r_{31}(X_n - T_x) + r_{32}(Y_n - T_y) + r_{33}(Z_n - T_z)} ,
\end{aligned}
\tag{7}
$$

where $f$ is focal length. $X_n$, $Y_n$ and $Z_n$ are object coordinates in the ground coordinate system at time $(t-1)$. $x_n$ and $y_n$ are left image coordinates at time $t$. $T_x$, $T_y$, $T_z$ are Translation element. $r_{11\sim33}$ are $3 \times 3$ rotation matrix elements. The $n$ is 1 to the number of corresponding points.

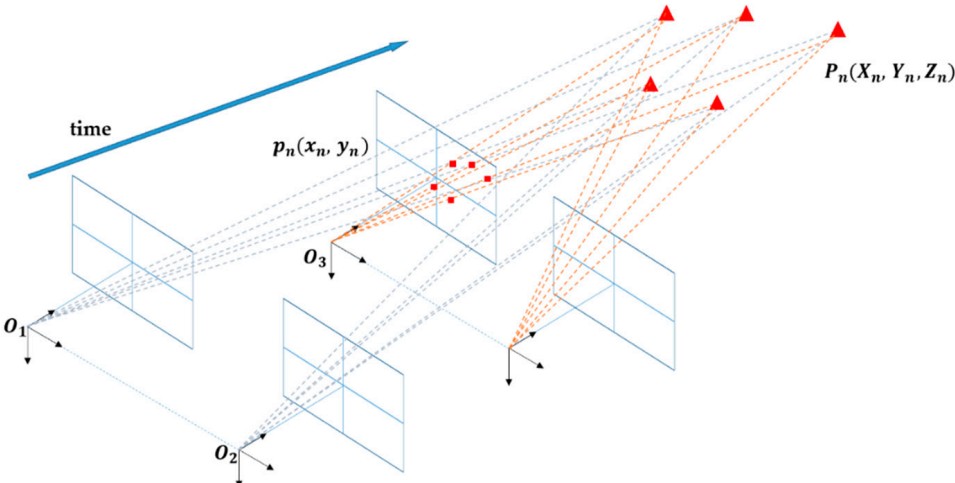

**Figure 7.** Absolute orientation in mobile mapping system (MMS).

We set Equation (8) by differentiating partially Equation (7) for the unknown. Then, we estimated geometric elements through the iterative least squares method.

$$
\begin{bmatrix}
\frac{\delta F}{\delta T_x} & \frac{\delta F}{\delta T_y} & \frac{\delta F}{\delta T_z} & \frac{\delta F}{\delta \omega} & \frac{\delta F}{\delta p} & \frac{\delta F}{\delta k} \\
\frac{\delta G}{\delta T_x} & \frac{\delta G}{\delta T_y} & \frac{\delta G}{\delta T_z} & \frac{\delta G}{\delta \omega} & \frac{\delta G}{\delta p} & \frac{\delta G}{\delta k} \\
\vdots & \vdots & \vdots & \vdots & \vdots & \vdots
\end{bmatrix}
\begin{bmatrix}
\Delta T_x \\ \Delta T_y \\ \Delta T_z \\ \Delta \omega \\ \Delta p \\ \Delta k
\end{bmatrix}
=
\begin{bmatrix}
-F_0 \\ -G_0 \\ \vdots
\end{bmatrix}, \tag{8}
$$

The estimated geometric elements mean the pose of $O_3$ by $O_1$. Therefore, we converted it to a $4 \times 4$ transformation matrix and then accumulated the pose by multiplying each result.

## 3. Results

We performed experiments with ten sequences in the KITTI dataset. The specifications of the computer used were Windows 10 64 bit, CPU i5-6600 3.30 GHz, RAM 16 GB, and the experiment was performed in visual studio 2013, Microsoft product in the United States. This section shows the results of corresponding point optimization and pose estimation. Then, it described the performance of the proposed method.

### 3.1. Corresponding Point Optimization Result

In the figure above, the turquoise lines indicate the feature motion vector. Figures 8 and 9 show the feature tracking results with and without optimization. In the figures, circles indicate the feature motion vector for the moving vehicle. As shown, it can be seen that this feature has a different motion vector from the surrounding points. We confirmed that the abnormal features indicated by circles were eliminated through optimization.

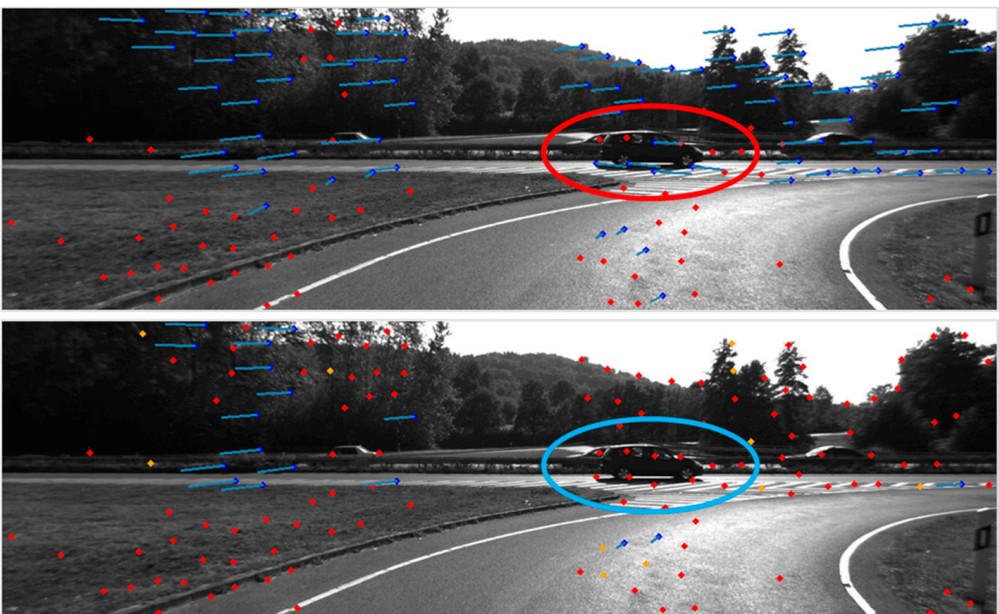

**Figure 8.** Top: before optimization (top image). Bottom: after optimization (sequence 01; frame 02).

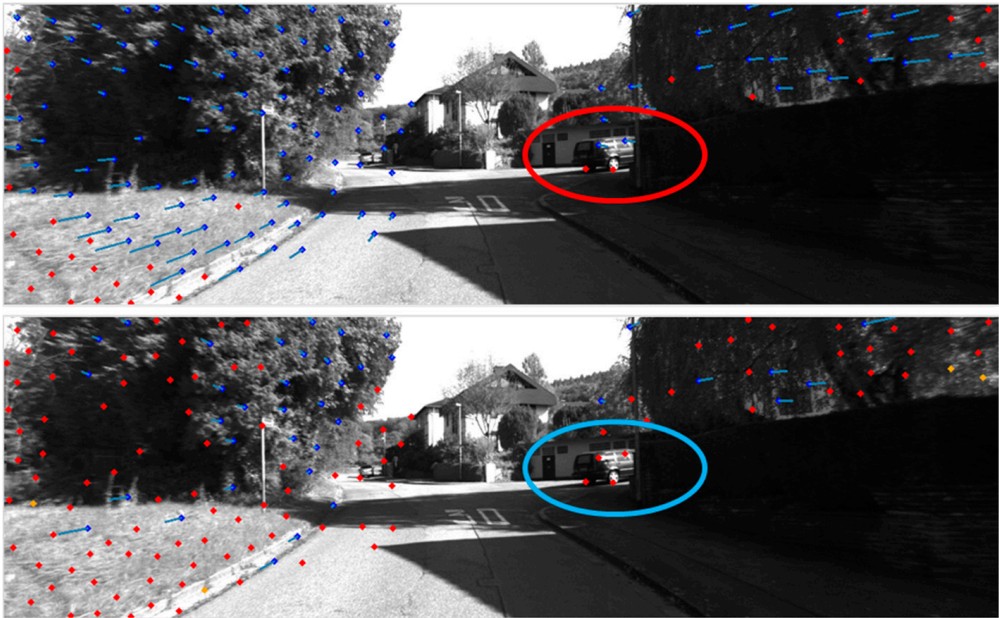

**Figure 9.** Top: before optimization (top image). Bottom: after optimization (sequence 03; frame 233).

Figures 10–12 shows the photogrammetric feature optimization result within sequence 09. In the top images, the red point indicates an outlier removed by vision-based optimization, and the orange point indicates an outlier removed by photogrammetry-based optimization. We confirmed that the features for the moving object were removed in two steps.

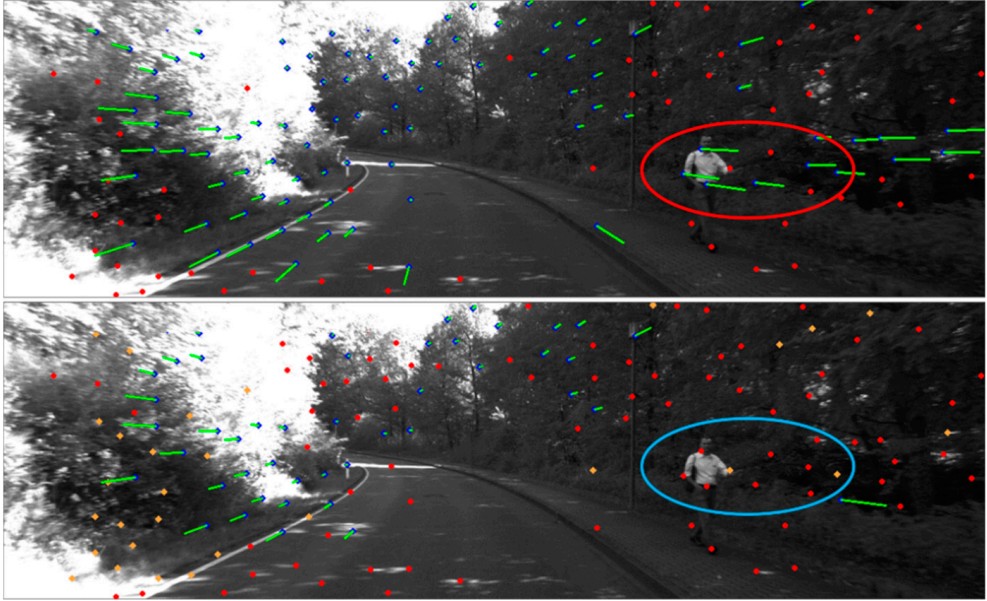

**Figure 10.** Top: before optimization (top image). Bottom: after optimization (sequence 09; frame 60).

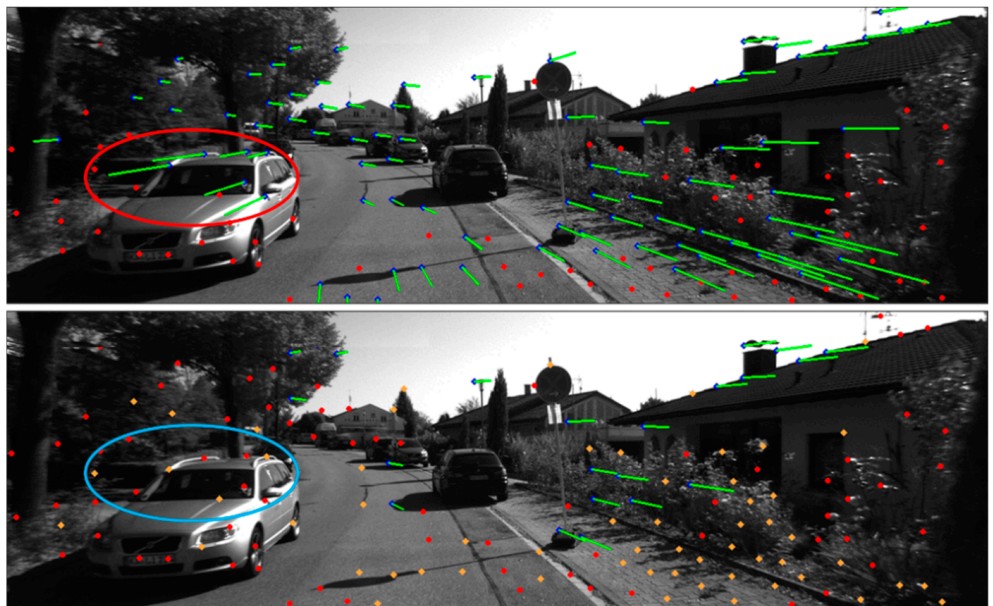

**Figure 11.** Top: before optimization (top image). Bottom: after optimization (sequence 09; frame 526).

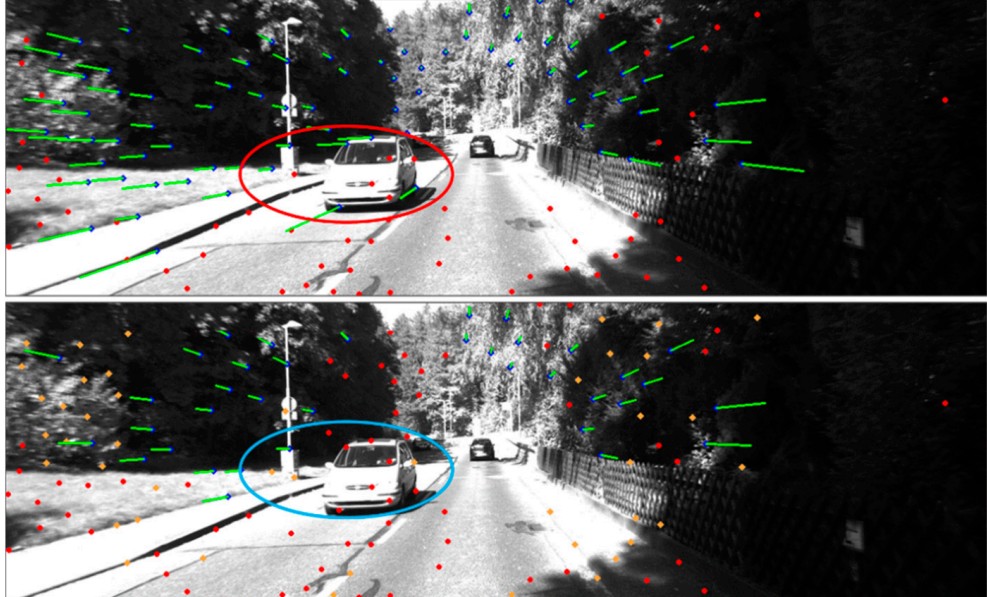

**Figure 12.** Top: before optimization (top image). Bottom: after optimization (sequence 09; frame 1436).

Figure 13 and Table 3 show the results with or without photogrammetric feature optimization. The rotation error rate decreased by 8.9383 deg/m and the translation error rate decreased by 0.0176%. As shown in Figures 10–12, we confirmed that the accuracy was improved by not using dynamic objects as features.

**Table 3.** Accuracy with or without optimization.

|  | Rotation Error Rate (deg/m) | Translation Error Rate (%) |
|---|---|---|
| Before optimization | 12.8637 | 0.0354 |
| After optimization | 3.9254 | 0.0178 |

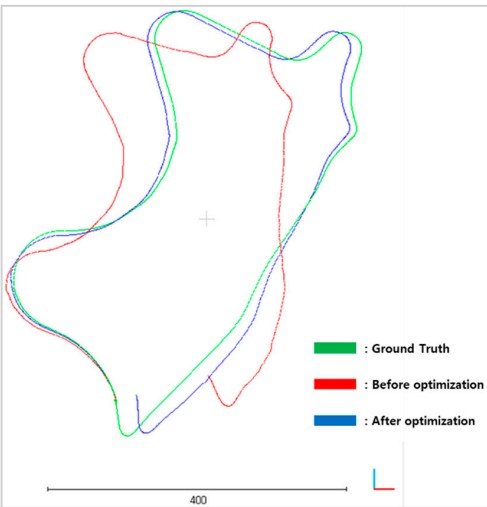

**Figure 13.** Estimated trajectory based on whether or not optimization is performed.

*3.2. Pose Estimation Result for Three Cases*

We checked the trajectory result by the path shape. We experimented with the sequence acquired in the area with fewer curves, a large number of curves, and a sharp curve. In Figure 14, the red line indicates the ground truth provided by KITTI, and the blue line indicates the trajectory estimated by the proposed method. As shown, the trajectory was more sensitive to the number of curve appearances rather than the degree of the curve. For three cases, the rotation error rate was 0.0156 deg/m, the translation error rate 2.8727%, and the processing time per frame was 0.0313 s on average.

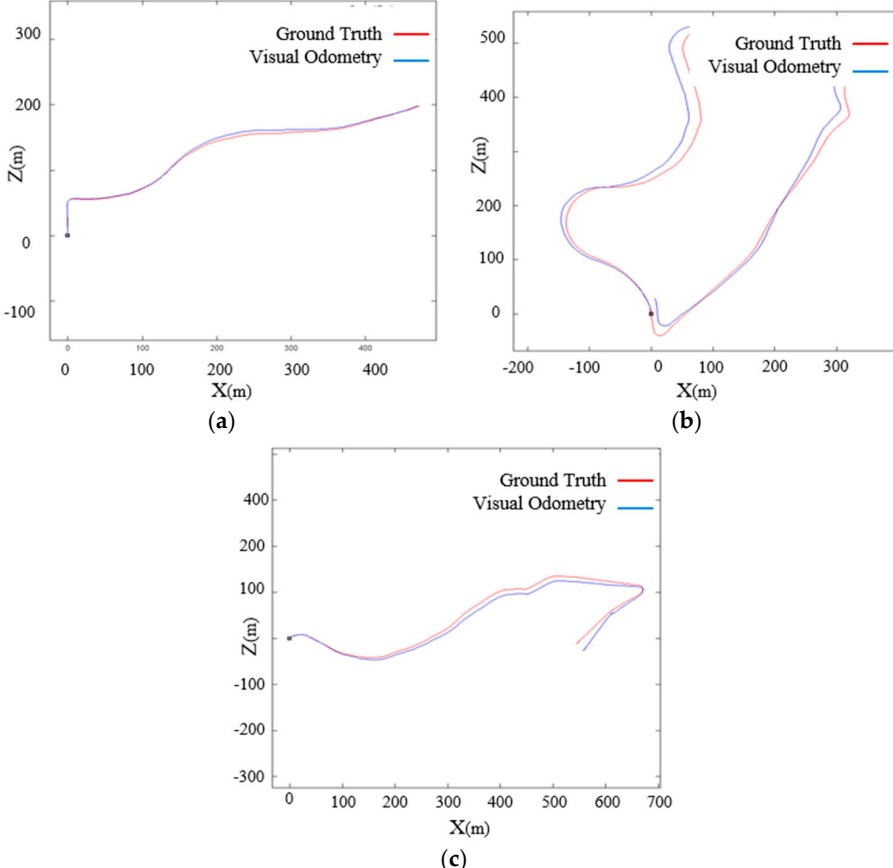

**Figure 14.** (**a**) Estimated result with fewer curves area; (**b**) The result with a high number of curves area; (**c**) The result with sharp curve.

### 3.3. Estimation Results for Ten Sequences in KITTI Dataset

Finally, we experimented with 00 to 11 (except 01) sequences provided by KITTI. Figure 15 is a graph showing the error occurrence per mileage with sequence 00, where (a) is about rotation and (b) is about translation. Figures 16–24 are graphs for sequences 01 to 11.

For ten sequences, the average rotation error was 0.0175 deg/m, the average translation error was 3.5520%, and the processing time per frame was 0.0554 s on average.

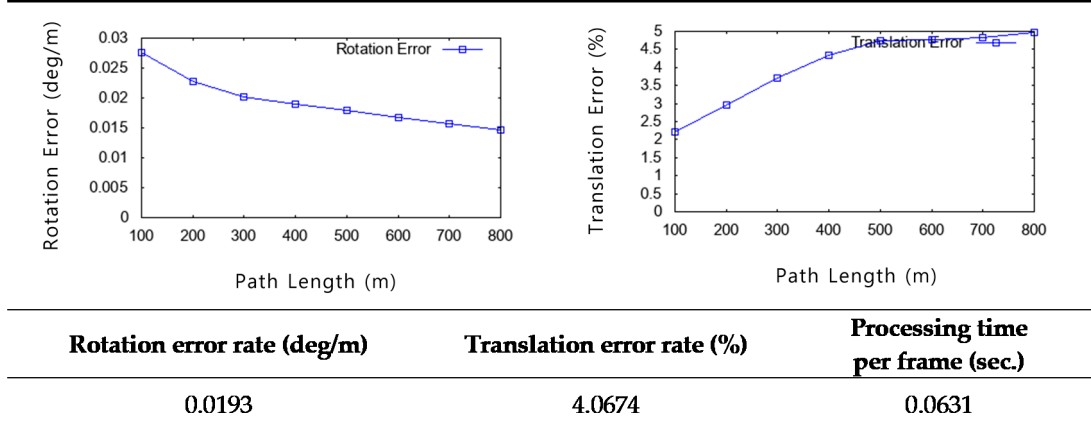

| Rotation error rate (deg/m) | Translation error rate (%) | Processing time per frame (sec.) |
| --- | --- | --- |
| 0.0193 | 4.0674 | 0.0631 |

**Figure 15.** Graph of error over mileage for sequence 00.

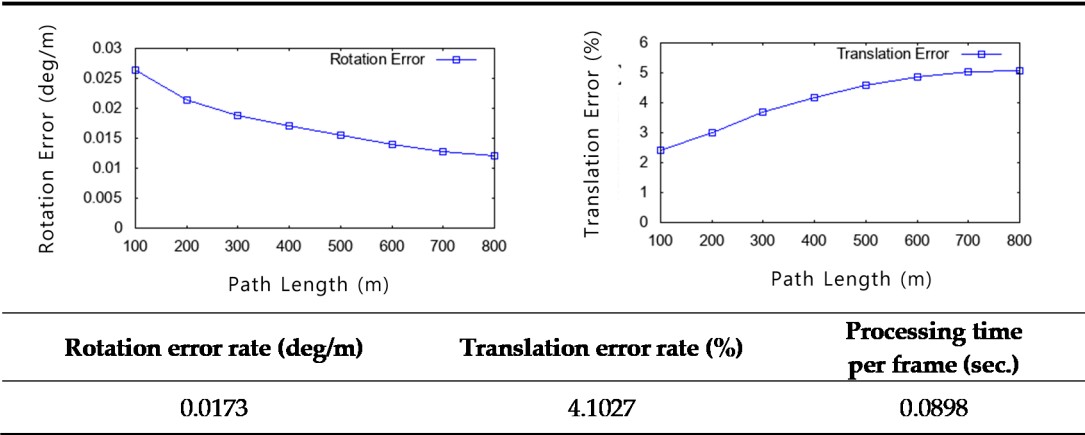

| Rotation error rate (deg/m) | Translation error rate (%) | Processing time per frame (sec.) |
| --- | --- | --- |
| 0.0173 | 4.1027 | 0.0898 |

**Figure 16.** Graph of error over mileage for sequence 02.

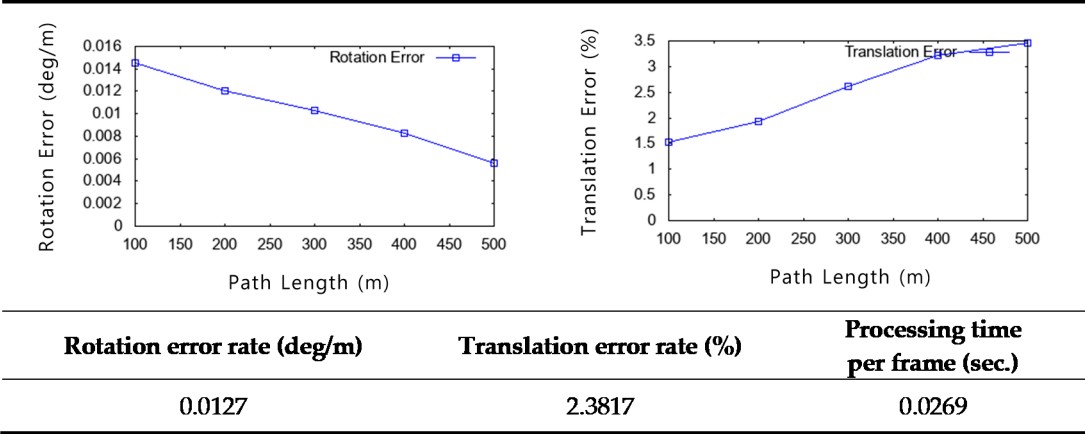

| Rotation error rate (deg/m) | Translation error rate (%) | Processing time per frame (sec.) |
| --- | --- | --- |
| 0.0127 | 2.3817 | 0.0269 |

**Figure 17.** Graph of error over mileage for sequence 03.

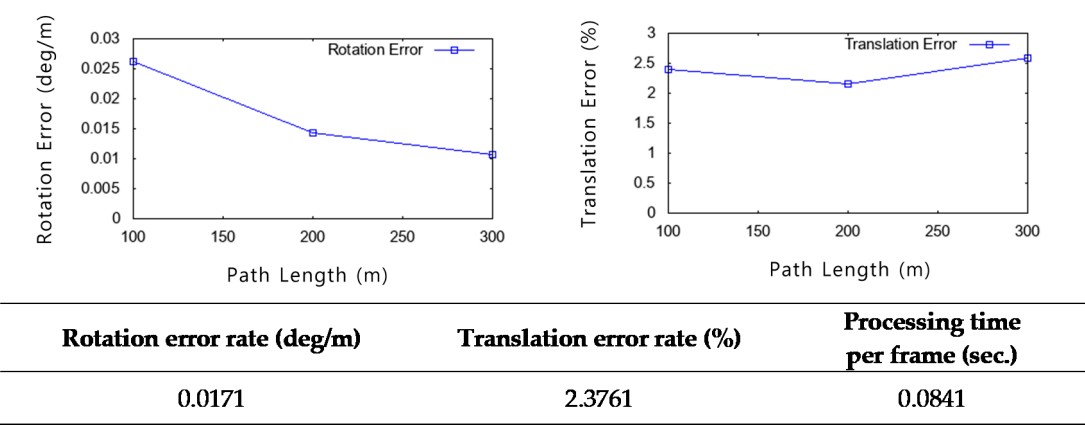

| Rotation error rate (deg/m) | Translation error rate (%) | Processing time per frame (sec.) |
| --- | --- | --- |
| 0.0171 | 2.3761 | 0.0841 |

**Figure 18.** Graph of error over mileage for sequence 04.

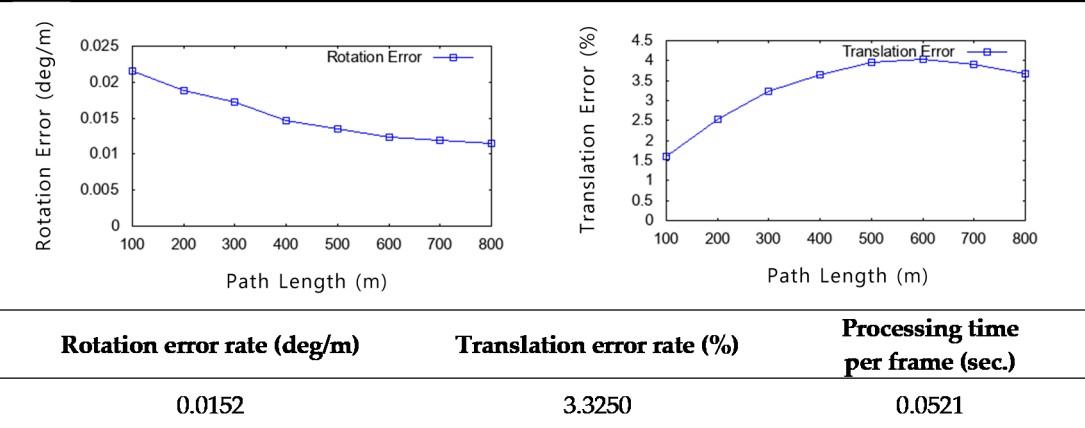

| Rotation error rate (deg/m) | Translation error rate (%) | Processing time per frame (sec.) |
| --- | --- | --- |
| 0.0152 | 3.3250 | 0.0521 |

**Figure 19.** Graph of error over mileage for sequence 05.

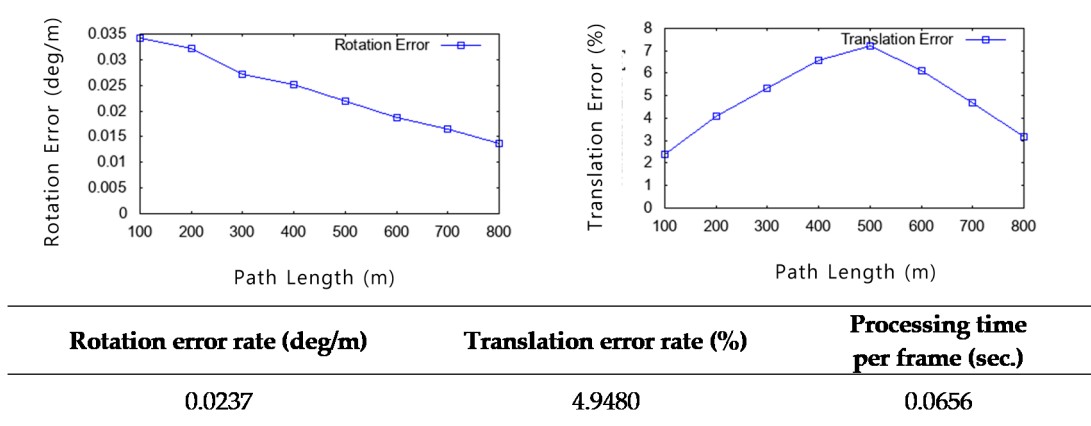

| Rotation error rate (deg/m) | Translation error rate (%) | Processing time per frame (sec.) |
| --- | --- | --- |
| 0.0237 | 4.9480 | 0.0656 |

**Figure 20.** Graph of error over mileage for sequence 06.

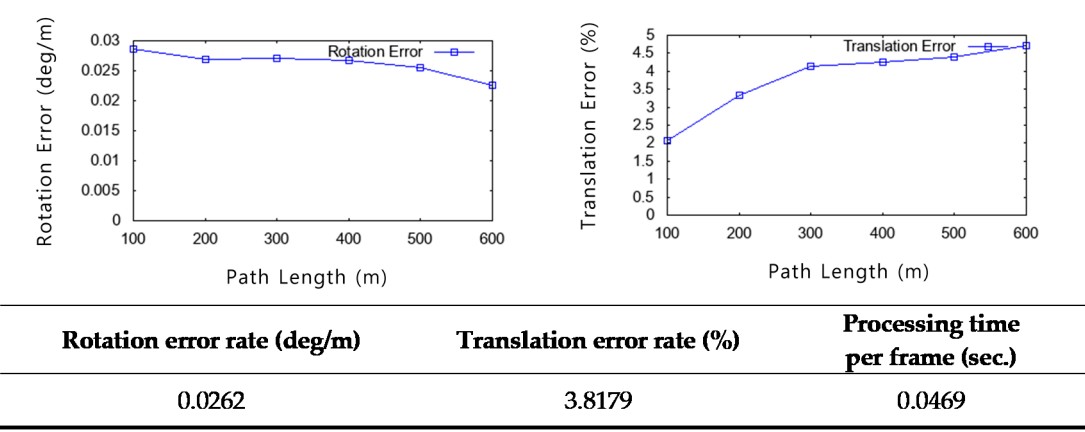

| Rotation error rate (deg/m) | Translation error rate (%) | Processing time per frame (sec.) |
|:---:|:---:|:---:|
| 0.0262 | 3.8179 | 0.0469 |

**Figure 21.** Graph of error over mileage for sequence 07.

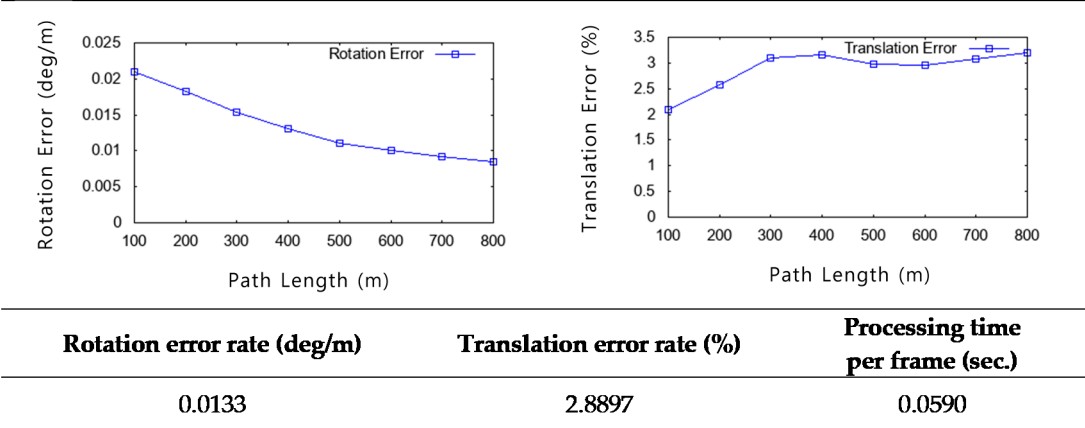

| Rotation error rate (deg/m) | Translation error rate (%) | Processing time per frame (sec.) |
|:---:|:---:|:---:|
| 0.0133 | 2.8897 | 0.0590 |

**Figure 22.** Graph of error over mileage for sequence 08.

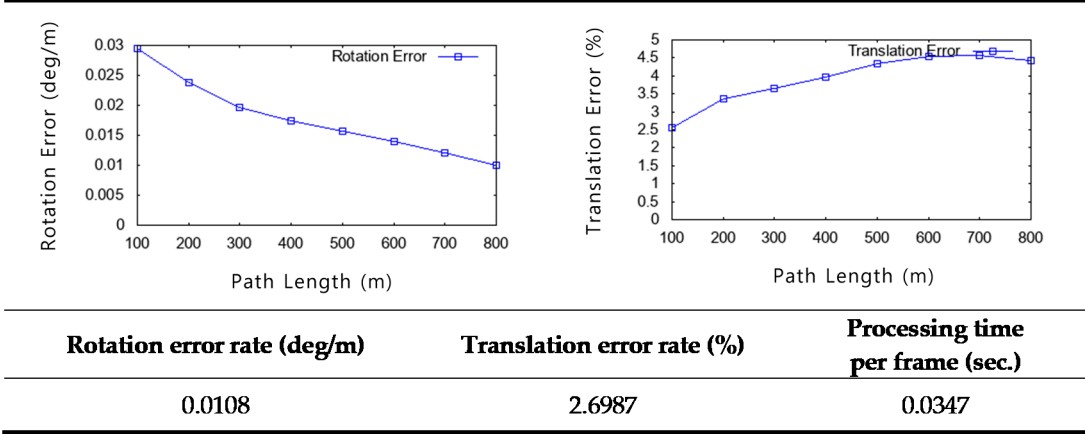

| Rotation error rate (deg/m) | Translation error rate (%) | Processing time per frame (sec.) |
|:---:|:---:|:---:|
| 0.0108 | 2.6987 | 0.0347 |

**Figure 23.** Graph of error over mileage for sequence 09.

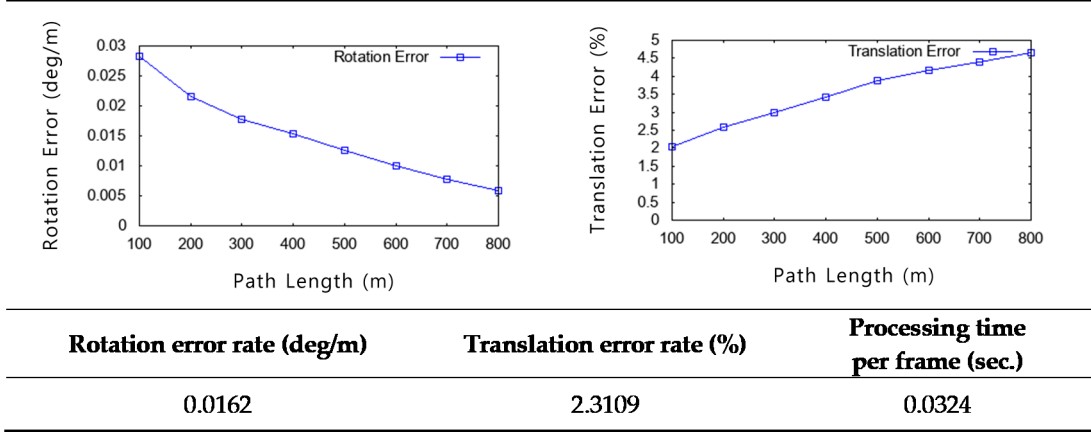

| Rotation error rate (deg/m) | Translation error rate (%) | Processing time per frame (sec.) |
|---|---|---|
| 0.0162 | 2.3109 | 0.0324 |

**Figure 24.** Graph of error over mileage for sequence 10.

## 4. Discussion

Through the comparison of Figures 8 and 9, we could see that the features on the moving object were eliminated by the proposed optimization scheme. In order to confirm the effectiveness of the optimization, the experiment was performed with sequence 09. Through the comparison of Figures 10–12, the features on moving objects were eliminated, and the rotation error rate decreased by 8.9383 deg/m, while the translation error rate decreased by 0.0176%. Then, we experimented with three different zones with different numbers and degrees of curves. The rotation error rate was smaller than the translation error rate. Also, we observed that the error generally occurred in the curved road rather than the straight road. In Table 4, the average processing time per frame was 0.0313 s. For ten sequences provided by KITTI, the average rotation error was 0.0175 deg/m, translation error was 3.5520%, and the running time per frame was 0.0554 s. The rotation error tended to decrease with the moving distance, but the translation error tended to increase. Through all experiments shown, we confirmed that the proposed feature optimization scheme worked successfully and that real-time processing was possible.

**Table 4.** Pose estimation accuracy for three cases.

| Sequence | Rotation Error Rate (deg/m) | Translation Error Rate (%) | Processing Time Per Frame (s) |
|---|---|---|---|
| (a) | 0.0127 | 2.3817 | 0.0269 |
| (b) | 0.0108 | 2.6987 | 0.0347 |
| (c) | 0.0162 | 2.3109 | 0.0324 |
| Average | 0.0132 | 2.4638 | 0.0313 |

Our research is ongoing and the performance shown here needs further improvements, particularly compared to known optimal algorithms. For example, SOFT2 [4] achieved a rotation error of 0.014 deg/m, a translation error of 0.65%, and a processing time of 0.1 s/frame. Nevertheless, our results support our intention of using photogrammetric analysis as an alternative outlier removal method. We showed that the proposed photogrammetric processing could enable successful outlier removal and that real-time processing was feasible even with photogrammetric iterative estimations. It is notable that we adopted the concept of circular matching proposed in CFORB [8] and enhanced its performance by photogrammetric optimizations. CFORB achieved a rotation error of 0.0107 deg/m, a translation error of 3.73%, and a processing time of 0.9 s/frame [8]. CFORB performed slightly better in translation errors compared to ours. This is because CFORB utilized the time-consuming RANSAC process repeatedly by 50 loops. One can check this by the large processing time of CFORB. However, such extensive RANSAC-based outlier removal may not bring accurate pose estimation, which is supported by the superior angular estimation performance by our method. The proposed

photogrammetric processing method could effectively remove outliers and estimate the pose correctly within a very small processing time.

## 5. Conclusions

Favorable feature extraction and outlier removal are key to visual odometry techniques. In this paper, we proposed photogrammetric feature optimization applicable to stereo odometry. Using the estimated poses of previous frames, we repeated the process of projecting and re-projecting the corresponding points extracted from the current frame onto the previous ones. Then, we removed the outliers by confirming the projection and re-projection errors. In addition, we optimized the feature on the new input image through multi-image filtering. Through the experiments, we were able to confirm the applicability of the proposed photogrammetric feature optimization process to stereo visual odometry technology.

We need to enhance the performance of the proposed optimization process further as there were some remaining outliers after optimization. In this paper, we considered photogrammetric analysis between a stereo pair of current and previous frames. We need to accumulate the results of incoming frames to remove outliers with better accuracy. Also, we need to consider preprocessing multiple stereo pairs of previous frames to generate a list of reference features for incoming frames. The major contribution of this paper is that we showed the feasibility of real-time outlier removal by photogrammetric analysis.

**Author Contributions:** All authors contributed in the developing method and editing of the paper. S.-J.Y. is the main author who designed whole experiments and wrote the manuscript.

**Funding:** This work was supported by the National Research Foundation of Korea (NRF) funded by the Korea government (MSIP) (No. NRF-2016R1A2B4013017).

**Conflicts of Interest:** The authors declare no conflict of interest.

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
