# Peer review of "Development of Stereo Visual Odometry Based on Photogrammetric Feature Optimization"

_remotesensing, doi:10.3390/rs11010067_

Round 1

Reviewer 1 Report

See attached file.

Author Response

Thank you for your efforts.

Reviewer 2 Report

The paper presents a methodology for the improvement of the results obtained in Stereo Visual Odometry through the photogrammetric treatment of existing features in the scenes. The subject is undoubtedly of great interest and great applicability, this method SVO being one of the most projected today.

The work is well structured, presenting a brief introduction, the proposed methodology, the data sets used, their results and the discussion and conclusions reached. However, the small number of references used in it is surprising, despite the fact that there is a large number of works on these topics, which is why authors are recommended more work in this line.
Although a methodological proposal is proposed, the terms of the same, as opposed to other previous schemes, are not sufficiently clear in the current wording of the work. The elimination of objects in motion in the scenes is a common practice, and in many cases photogrammetric techniques are used based on the available information, and by an analysis of differential speeds. It would be necessary to clarify what is specifically the novelty of the work compared to other methodologies that are already being used.
Finally, note that although there are different examples of the improvement of calculated trajectories (compared to the actual trajectory) by proposing the elimination of objects in the scene, or not, a comparison with other methods of improving the trajectory would have been advisable that are used today.

Author Response

Thank you for your efforts.

Reviewer 3 Report

- Moderate English revision is recommended.

- Introduction focuses on analyzing competing approaches on a cost basis only. A slight performance comparative analysis would be appreciated. What about LiDAR SLAM?

- The core development of the paper is adequate.

- Seems that outlier focus is given to moving vehicles or persons. What about points in vegetation (which seem to be quite a few)? how can you measure the goodness of a match?

Author Response

Thank you for your efforts.

Round 2

Reviewer 2 Report

Once the new version of the work has been revised, it can be concluded that it does not include significant improvements that lead to the modification of my initial assessment. As the authors themselves acknowledge in their cover letter, there has not been a detailed analysis of other methodologies and on the other hand it is a methodology whose development is in process.

In particular, in my first review I indicated the interest of the work but also the need to attend to two basic aspects so that the work represented an adequate scientific contribution (contribution to knowledge) for its publication in Remote Sensing. These comments focused on:

1) The small number of used references in it is surprising, despite the fact that there is a large number of works on these topics, which is why authors are recommended more work in this line (only some references are included, but the authors need a deeper work in this aspect). My suggestion was not oriented only to include a greater number of references (that would be unimportant since the number does not give the quality) but a deeper analysis of the work done in that line.

2) Include comparisons with other methods that are currently being used for this same objective, in order to be able to analyze the improvement implied by the proposed methodology. This recommendation has not been addressed, and only some comments have been included, but not enough supported by objective data.

Author Response

The authors sincerely appreciate the time and efforts taken again to review the revised version of our manuscript.

Sungjoo
